# The Histone H3K27 Demethylase REF6 Is a Positive Regulator of Light-Initiated Seed Germination in Arabidopsis

**DOI:** 10.3390/cells12020295

**Published:** 2023-01-12

**Authors:** Yahan Wang, Dachuan Gu, Ling Deng, Chunmei He, Feng Zheng, Xuncheng Liu

**Affiliations:** 1Key Laboratory of South China Agricultural Plant Molecular Analysis and Genetic Improvement and Guangdong Provincial Key Laboratory of Applied Botany, South China Botanical Garden, Chinese Academy of Sciences, Guangzhou 510650, China; 2College of Life Sciences, University of Chinese Academy of Sciences, Beijing 100049, China

**Keywords:** histone methylation, histone demethylation, REF6, light-initiated seed germination

## Abstract

Seed germination is the first step in initiating a new life cycle in seed plants. Light is a major environmental factor affecting seed germination. Phytochrome B (phyB) is the primary photoreceptor promoting germination during the initial phase of imbibition. Post-translational histone methylation occurring at both lysine and arginine residues plays a crucial role in transcriptional regulation in plants. However, the role of histone lysine demethylation in light-initiated seed germination is not yet reported. Here, we identified that Relative of Early Flowering 6 (REF6)/Jumonji Domain-containing Protein 12 (JMJ12), a histone H3 lysine 27 (H3K27) demethylase, acts as a positive regulator of light-initiated seed germination. The loss of function of *REF6* in Arabidopsis inhibits phyB-dependent seed germination. Genome-wide RNA-sequencing analysis revealed that REF6 regulates about half of the light-responsive transcriptome in imbibed seeds, including genes related to multiple hormonal signaling pathways and cellular processes. Phenotypic analyses indicated that REF6 not only regulates seed germination through GA (gibberellin) and ABA (abscisic acid) processes but also depends on the auxin signaling pathway. Furthermore, REF6 directly binds to and decreases the histone H3K27me3 levels of auxin-signaling- and cell-wall-loosening-related genes, leading to the activated expression of these genes in imbibed seeds. Taken together, our study identifies REF6 as the first histone lysine demethylase required for light-initiated seed germination. Our work also reveals the important role of REF6-mediated histone H3K27 demethylation in transcriptional reprogramming in the light-initiated seed germination process.

## 1. Introduction

Seed germination, a key ecological and agronomic trait, is one of the most critical phases in the plant life cycle, determining when plants enter ecosystems [1]. Arabidopsis seeds consist of the embryo, which is surrounded by a single cell layer of endosperm, and the testa [2], and the germination process includes testa rupture and subsequent endosperm rupture [3,4]. Seed germination is precisely regulated by environmental cues and diverse endogenous plant hormones [5,6].

Light is an important environmental signal and energy source for plant life, regulating almost the entire life cycle of plants [7]. Red light (R) induces seed germination, and subsequent far-red (FR) light irradiation reverses this process in dark-imbibed seeds [8]. R/FR-light-convertible phytochromes are the main receptors of light-mediated seed germination [9]. Phytochromes are synthesized in a signaling-inactive state known as Pr. Upon R absorption, the Pr form is converted to a signaling-active form known as Pfr [10,11]. The Arabidopsis genome encodes five phytochromes, phyA to phyE [12]. phyA and phyB play major roles in light-initiated seed germination [13,14]. phyA promotes germination in response to very low fluence and high FR irradiation, whereas phyB induces germination in response to low red-light fluence. Due to low protein levels of phyA, germination predominantly relies on active phyB in imbibed seeds of Arabidopsis [13,14].

Phytochromes perceive a light signal and affect the transduction of plant hormone signals, such as GA (gibberellin), ABA (abscisic acid) and auxin, in plants [15]. It is well known that GA promotes seed germination, while ABA inhibits it. Red-light-activated phyB up-regulates the expression of GA biosynthesis genes, whereas it down-regulates the expression of ABA biosynthesis genes [16]. Auxin has also been implicated in regulating seed germination [1]. A low concentration of auxin promotes seed germination, whereas a high concentration of auxin inhibits seed germination [17]. It was reported that auxin efflux carriers *PIN1*, *PIN2*, *PIN3*, and *PIN7*, the auxin influx transporter *AUX1*, and many auxin-responsive genes are activated by red-light signals in imbibed seeds [18]. Apart from the effect of hormone signals, Arabidopsis seed germination involves sequential testa rupture, endosperm rupture, and embryo radicle protrusion. The induction of cell-wall-loosening enzymes, including Xyloglucan Endotransglucosylases/Hydrolases (XTHs), Expansins (EXPs) and Pectinesterases/Pectin Methylesterases (PMEs), plays an important role in the promotion of seed germination [1].

Light triggers the reprogramming of approximately 30% of the transcriptome in plants [19,20]. Accumulating evidence suggests that light-mediated transcriptional regulation involves dynamic histone modification changes [21,22,23]. As one of the most complex modifications, histone methylation occurs at both lysine and arginine residues. The lysine methylation of histones is associated with both transcriptional gene silencing and activation. Generally, histone H3 lysine 9 (H3K9) and H3K27 methylation is associated with gene repression, whereas H3K4 and H3K36 methylation is related to gene activation [24,25]. Reversible histone methylation and demethylation are catalyzed by histone methyltransferases (HMTs) and histone demethylases (HDMs), respectively [25]. In Arabidopsis, numerous studies have revealed the involvement of histone methylation in light-initiated seed germination. A mutation in EFS (Early Flowering in Short Days), an H3K4 and H3K36 methyltransferase, promotes seed germination even under phyB inactivation conditions [26]. EFS represses the expression of the seed germination regulator *PIF1* via H3K36 methylation [26]. SUVH5, a histone H3K9 methyltransferase, positively regulates light-mediated seed germination by repressing ABA-signaling-related genes and *DELAY OF GERMINATION* (*DOG*) via H3K9me2 in imbibed seeds [27]. Moreover, the loss of function of two histone arginine demethylases, *JMJ20* and *JMJ22*, results in a reduction in seed germination under phyB activation conditions [28]. JMJ20 and JMJ22 positively regulate seed germination by repressing the expression of two key GA biosynthesis genes, *GA3ox1* and *GA3ox2*, via the removal of histone arginine methylation [28]. Collectively, these works revealed that histone lysine methylation at H3K4, H3K9 and H3K36 residues and histone arginine methylation play an important role in the transcriptional regulation of light-initiated seed germination; however, the role of histone lysine demethylation in light-initiated seed germination remains largely unclear.

In the present work, we report that REF6, a histone H3K27 demethylase, plays a positive role in phyB-dependent seed germination. Transcriptomic analysis reveals that REF6 may act as a crucial component in light-mediated transcriptional reprogramming in imbibed seeds. Moreover, REF6 promotes light-initiated seed germination by directly activating auxin-signaling- and cell-wall-loosening-related genes via the removal of H3K27me3. Our findings suggest that REF6-mediated histone H3K27 demethylation plays an important role in transcriptional reprogramming in the light-initiated seed germination process.

## 2. Materials and Methods

### 2.1. Plant Materials

All Arabidopsis seeds used in this study are in the Col-0 background. The *ref6-1* [29] mutant was a kind gift from professor Xingliang Hou at the South China Botanical Garden. The *phyB-9* [30] mutant was obtained from TAIR. To construct *ProREF6-REF6-GFP* transgenic plants, the native promoter and full-length coding sequence of *REF6* were subcloned into a pCambia-1302 vector and transformed into the *ref6-1* mutant background. The seeds used for light-dependent germination were harvested in the same batch of plants grown at 22 °C under long days (16 h WL/8 h dark). Following harvesting, seeds were dried in an incubator at 22 °C for 30–35 days prior to germination assays.

### 2.2. phyB-Dependent Germination Assays

phyB-dependent seed germination assays were performed as described previously [31]. Briefly, seeds were surface-sterilized and plated on half-strength Murashige–Skoog (Sigma-Aldrich, St. Louis, MO, USA) agar plates containing 0.3% sucrose and 1% phytoagar (pH 5.7). The plates were placed in an illuminated incubator with white light (100 μmol m^−2^ s^−1^) at 22 °C. One hour after sterilization, seeds were irradiated with far-red light (3.8 μmol m^−2^ s^−1^) for 5 min (indicated as FR or phyB-off) or exposed to far-red light (3.8 μmol m^−2^ s^−1^) for 5 min followed by irradiation with red light (13.1 μmol m^−2^ s^−1^) for 5 min (referred as FR/R or phyB-on). The seeds were kept in the dark either for 24 h for the gene expression and ChIP analyses or for 4 days to calculate the germination rates. To examine the germination rates, at least 50 seeds were used for each experimental point, and three biological replicates were used for statistical analysis.

### 2.3. RNA Isolation and Quantitative RT-PCR (qRT-PCR) Analysis

After FR or FR/R treatment, seeds were incubated in the dark at 22 °C for 24 h. About 0.1 g of seeds for each sample was used for the assay. Total RNA was extracted with TRIZOL Reagent (Invitrogen, Shanghai, China) according to the manufacture’s protocol. After DNAse I treatment, first-strand cDNA was synthesized using 2 μg of total RNA according to the manufacturer’s instruction of the TransScript One-Step gDNA Removal and cDNA Synthesis Super Mix Kit (TransGen, Beijing, China). qRT-PCR assays were performed by using SYBR Green Mix (Bio-Rad) in an ABI7500 Real-Time PCR System (Applied Biosystems, Waltham, MA, USA). Each sample was quantified at least in triplicate. All PCR reactions were normalized using the Ct value corresponding to the reference gene *PP2A*. The relative expression levels of target genes were calculated with the formula 2^-ddCt^. Values represented the average of three biological replicates. The primer pairs for qRT-PCR are listed in Appendix A.

### 2.4. RNA-Sequencing (RNA-seq) Analysis

For high throughput RNA-seq analysis, total RNA was extracted as described above, and an mRNA-seq library was constructed by using an mRNA Seq Kit (Illumina). RNA-seq was performed by Millennium Genomics in triplicates. High-quality clean reads were obtained after removing adaptor sequences, ambiguous reads (‘N’ > 10%), and low-quality reads (i.e., more than 50% of bases in a read had a quality value Q ≤ 5). Then, the clean reads were mapped to the Arabidopsis genome TAIR10 using HISAT2 software with default parameters [32]. Cuffdiff (http://cole-trapnell-lab.github.io/cufflinks/, accessed on 12 January 2021) was applied to detect differentially expressed genes (DEGs) [33]. Genes with fold changes >1.5 with statistical significance (*p* < 0.05) were selected. GO (Gene Ontology) analyses of DEGs were performed with Metascape software (http://metascape.org, accessed on 12 October 2022) with a cutoff of *p* < 0.05 and a minimum overlap of 3. The regulation trends of DEGs were visualized by using heatmaps made in Heml [34]. 

### 2.5. ChIP Assays

ChIP assays were performed as previously described [35] with minor modifications. About 0.3 g of post-harvest seeds was used for each sample. After FR/R treatment, the seeds were incubated in the dark for 24 h and then fixed in crosslinking solution (1% formaldehyde) under vacuum for 1 h. Glycine was added to a final concentration of 0.125 M and incubated for another 5 min to terminate crosslinking. The chromatin was extracted and suspended with 600 uL of lysis buffer and then sheared to an average length of 500 bp by sonication. After centrifugation, the supernatant was diluted and then immunoprecipitated overnight at 4 °C with specific antibodies, including anti-H3K27me3 (Catalog no. 07-449, Millipore, MA, USA) or anti-GFP beads (Catalog no. KTSM1334, KTSM-life, Shenzhen, China). After washing, the DNA–protein complex was eluted, and crosslinking was then reversed by incubating the samples with 5 M NaCl for 6 h at 65 °C. The amount of immunoprecipitated DNA fragments was determined by quantitative PCR and normalized to the internal control *ACTIN2.* Each sample was quantified at least in triplicate. All PCR reactions were normalized using the Ct value corresponding to the internal control gene *ACTIN2*. The fold change was calculated with the formula 2^-ddCt^. The primer pairs for ChIP-qPCR are listed in Appendix A.

### 2.6. Statistical Analysis

Statistical differences were assessed by Student’s *t*-test or one-way analysis of variance (ANOVA) using SPSS 13.0. Values of *p* < 0.05 and *p* < 0.01 were considered to be statistically significant and highly significant, respectively (compared with the wild type). Data are presented as mean ± SD.

## 3. Results

### 3.1. REF6 Is a Positive Regulator in phyB-Mediated Seed Germination

phyB is the major photoreceptor responsible for promoting seed germination [13]. To examine whether REF6 acts in the phyB-mediated seed germination process, the phenotype of *ref6-1* [29], a knockout mutant of *REF6*, was analyzed. For phyB-dependent seed germination assays, the seeds were irradiated with far-red light pulses (phyB-off, referred to as FR) or far-red light followed by red light (phyB-on, indicated as FR/R) and subsequently kept in the dark for 4 days [36] (Figure 1A). Seeds stored at 22 °C for 30–35 days after harvesting were used for germination assays. Under FR conditions, both wild-type (Col-0) and *ref6-1* seeds failed to germinate (Figure 1B,C). Four days after FR/R treatment, about 90% of Col-0 seeds germinated, while only about 60% of *ref6-1* seeds germinated (Figure 1B,C). In addition, under continuous white-light (WL) conditions, almost all of the Col-0 and *ref6-1* seeds germinated (Figure 1B,C). Collectively, the above data suggest that REF6 is a positive regulator of phyB-mediated seed germination in Arabidopsis.

To examine the genetic relationship between REF6 and phyB in the phyB-dependent seed germination process, we generated a *ref6 phyB* double mutant by crossing *ref6-1* and *phyB-9* [30]. Under phyB-off conditions, the seeds of Col-0, *ref6-1*, *phyB-9* and *ref6 phyB* failed to germinate (Figure 2A,B). Under phyB-on conditions, the seeds of the *ref6 phyB* double mutant displayed a similar germination pattern to the *phyB-9* single mutant (Figure 2C). Furthermore, we showed that the expression of *REF6* was up-regulated 3 and 6 h after FR/R treatment, whereas it was down-regulated 3, 6, 12 and 24 h after FR treatment (Appendix A). Collectively, these data indicate that REF6 may act downstream of phyB in the light-regulated seed germination process.

### 3.2. Genome-Wide Analysis of REF6-Regulated Transcriptome during Seed Germination

To further examine the regulatory role of REF6 in phyB-mediated seed germination, we analyzed the REF6-regulated transcriptome under FR/R conditions using RNA-sequencing (RNA-seq) assays. After FR/R treatment, seeds of Col-0 and *ref6-1* were kept in the dark for 24 h and then harvested for RNA extraction and cDNA library construction. To obtain reliable results, three independent biological replicates were prepared for high-throughput sequencing. A cutoff of 1.5-fold change with statistical significance (*p*-value < 0.05) was set to identify differentially expressed genes. 

Compared with Col-0, 1628 genes were up-regulated whereas 1511 genes were down-regulated in the *ref6-1* mutant, suggesting that REF6 may act as both a transcription repressor and activator in the light-initiated seed germination process (Appendix A). Gene Ontology (GO) analysis revealed that the up-regulated genes in *ref6-1* were preferentially associated with response to water, lipid storage, response to toxic substance, secondary metabolic process, seed maturation and response to abscisic acid (Figure 3A). In contrast, the down-regulated genes in *ref6-1* were mainly enriched in response to karrikin, cell wall organization or biogenesis, transmembrane transport, response to chitin, response to light stimulus, photosynthesis and cell wall modification (Figure 3B). We further showed that REF6 activates or represses a large subset of genes related to multiple key biological processes, including plant hormone, development, environment and cellular function (Figure 4A,B). Collectively, the RNA-seq data suggest that REF6 integrates both internal and external factors in the regulation of seed germination in Arabidopsis.

Previous RNA-seq analysis revealed that 2069 genes were differentially expressed in imbibed seeds under phyB activation conditions (FR/R) compared with those under phyB activation conditions (FR) (referred to as light-regulated genes) [18]. Heatmap and Venn diagram analyses indicated that about half of the light-regulated genes (919 of 2069) were also regulated by REF6 (Figure 3C,D, Appendix A). Interestingly, among these co-regulated genes, 869 genes (94.5%) were co-repressed or co-activated by REF6 and light, whereas only 50 genes (5.5%) were differentially regulated (Figure 3C,D, Appendix A), which implies that REF6 may act as a crucial component in light-mediated transcriptional reprogramming in imbibed seeds.

### 3.3. Seed Germination Phenotype of Ref6 Mutant in Response to ABA, GA and Auxin

RNA-seq analysis revealed that a large set of genes related to plant hormone signals, including GA, ABA and auxin, were either activated or repressed by REF6 in imbibed seeds (Figure 4). To further investigate the role of REF6 in hormone-mediated seed germination, we detected the germination rates of the *ref6* mutant in response to ABA, GA and auxin in a light-dependent manner.

We first tested the effects of GA and its inhibitor paclobutrazol (PAC) on the seed germination of *ref6-1*. Under phyB-on conditions, the 1 µM GA treatment markedly increased the germination rate of *ref6-1* (from 34% to 52%). Interestingly, the germination rate did not further increase even when supplemented with 10 µM GA (Figure 5A). Under phyB-off conditions, 5 and 10 µM GA treatments remarkably increased the germination rate of Col-0 seeds, whereas they had no obvious effect on *ref6-1* (Figure 5A). A GA sensitivity analysis indicated that *ref6* seeds were more sensitive to GA under phyB-on conditions, while they were less sensitive to GA under phyB-off conditions compared to Col-0 (Figure 5A). Furthermore, we showed that the addition of 0.1, 1 and 5 µM PAC significantly decreased the germination rates of both Col-0 and *ref6-1* under phyB-on conditions (Figure 5B). In addition, neither Col-0 nor *ref6-1* seeds germinated under phyB-off conditions after being supplemented with different concentrations of PAC (Figure 5B). These data suggest that REF6 may activate seed germination not only through GA but also through other pathways.

We next investigated the effect of ABA treatment on the seed germination of *ref6-1* in light-dependent conditions. Similar to Col-0, the addition of 0.1, 1, 5 and 10 µM ABA notably restrained the germination of *ref6-1* (Figure 5C) under phyB-on conditions, whereas both Col-0 and *ref6-1* failed to germinate under phyB-off conditions (Figure 5C), which indicates that REF6 regulates light-dependent seed germination via an ABA signaling process.

Finally, we examined the germination rates of *ref6-1* seeds in response to auxin. 1-Naphthylacetic acid (NAA), a synthetic auxin analog, was added to the medium for analysis. Under phyB-on conditions, low levels (0.1 and 0.5 µM) of NAA showed no obvious effect on the germination of Col-0; in contrast, the addition of 0.1 µM NAA significantly promoted the germination of *ref6-1*. On the other hand, a high level (10 µM) of NAA induced a slight decrease in the germination of Col-0, whereas it greatly reduced the germination rate of *ref6-1* seeds (Figure 6A,C). In addition, NAA treatments had no obvious effect on the germination of Col-0 and *ref6-1* seeds under phyB-off conditions (Figure 6B,D). The above data reveal that REF6 may promote seed germination largely via the auxin signaling pathway. 

Collectively, our phenotypic analyses suggest that REF6 regulation of seed germination not only occurs through the GA and ABA signaling pathways but also greatly depends on the auxin pathway.

### 3.4. REF6 Activates the Expression of Auxin-Signaling- and Cell-Wall-Loosening-Related Genes

As reported, REF6 is a plant-specific H3K27 demethylase that activates the expression of target genes via histone H3K27 demethylation [37,38]. Therefore, we focused on the genes activated by REF6 in imbibed seeds. RNA-seq analysis indicated that a number of auxin signaling components, including the auxin biosynthesis gene *YUC3*, the auxin efflux carriers *PIN1*, *PIN3* and *PIN7*, the auxin influx carrier *LAX1*, the auxin efflux transmembrane transporter *ABCB4* and the auxin-responsive genes *IAA18* and *SAUR51* were activated by REF6 in imbibed seeds (Figure 4A and Appendix A). Quantitative RT-PCR analysis further confirmed that the expression levels of *IAA8*, *PIN1*, *PIN3*, *PIN7*, *ABCB4* and *SAUR51* were significantly down-regulated in the *ref6* mutant compared with Col-0 under FR/R conditions (Figure 7A).

The rupture of endosperm is the key and last step controlling the process of seed germination, which is closely associated with cell wall remodeling and organization [3,4]. XTHs, EXPs and PMEs are key enzymes involved in cell wall loosening and expansion [2,39]. Our transcriptome data demonstrated that a large number of gene clusters, including *XTH9, XTH24*, *EXP3, EXPA9, EXPA10, EXPA1, EXPA15* and *PME1*, were activated by REF6 in imbibed seeds (Figure 4A and Appendix A). Further qRT-PCR analysis confirmed that the expression levels of *XTH9*, *XTH24*, *EXPA1*, *EXPA9*, *EXPA15* and *PME1* were significantly down-regulated in the *ref6-1* mutant compared with the wild type (Figure 7B). Collectively, the above data reveal that REF6 may promote phyB-dependent seed germination by activating auxin-signaling- and cell-wall-loosening-related genes in imbibed seeds.

### 3.5. REF6 Directly Targets Auxin-Signaling- and Cell-Wall-Loosening-Related Genes

To investigate whether these auxin-signaling- and cell-wall-loosening-related genes are direct targets of REF6 in imbibed seeds, transgenic lines expressing *REF6-GFP* in the *ref6-1* background were generated for chromatin immunoprecipitation (ChIP) assays (Appendix A). The expression of REF6-GFP could rescue the germination defect of *ref6-1* seeds under phyB-on conditions (Appendix A), which suggests that REF6-GFP is functional in vivo. Previous works reported that REF6 recognizes a specific CTCTGYTY motif in the Arabidopsis genome [37,38]. We further identified that auxin-signaling-related genes, such as *IAA18*, *PIN1*, *SAUR51*, *ABCB4* and *YUC3*, and cell-wall-loosening-related genes, including *XTH24*, *EXPA9*, *EXPA10*, *EXPA15* and *GH3.15*, harbor this motif in either their promoters or gene-body regions (Figure 8A). ChIP-qPCR analysis showed that REF6 is associated with the CTCTGYTY motif-containing regions of these genes in vivo (Figure 8B,C). The above data suggest that REF6 directly targets auxin-signaling- and cell-wall-loosening-related genes to promote their expression in imbibed seeds.

### 3.6. REF6 Activates Target Genes Expression via Removal of Histone H3K27me3

Previous studies reported that REF6 is a histone H3K27 demethylase and contributes to gene activation [37,38,40]. In Arabidopsis, the H3K27me3 mark preferentially localizes to the transcribed regions of genes [41]. Therefore, we detected the levels of H3K27me3 in the transcription start sites (TSSs) and first exons of the auxin-signaling- and cell-wall-loosening-related genes in Col-0 and *ref6* seeds through ChIP assays (Figure 9A).

After FR/R treatment, Col-0 and *ref6-1* seeds were kept in the dark for 24 h and then harvested for analysis. Compared with Col-0, the H3K27me3 levels in the TSSs and exon regions of *PIN3*, *PIN7*, *ABCB4* and *EXPA9* and the exon regions of *EXPA1* and *EXPA10* were significantly up-regulated in the *ref6* mutant (Figure 9B,C). These data suggest that REF6 activates the transcription of auxin-signaling- and cell-wall-loosening-related genes via the removal of histone H3K27m3 in imbibed seeds.

## 4. Discussion

Light-initiated seed germination involves dynamic transcriptional reprogramming [18]. Reversible histone modifications, including acetylation and methylation, play crucial roles in modulating the chromatin configuration and gene activity in plants [42]. In this study, we found that REF6, a histone H3K27me3 demethylase, plays a positive role in light-initiated seed germination in Arabidopsis. The loss of function of *REF6* leads to decreased seed germination under phyB-on conditions. Transcriptomic analysis revealed that REF6 affects about half of the light-responsive transcriptome in imbibed seeds, including gene clusters related to multiple hormone signaling pathways. Genetic and phenotypical analyses revealed that REF6 may act downstream of phyB and promote light-regulated seed germination through GA, ABA and auxin signaling pathways. Moreover, we showed that REF6 directly targets a subset of auxin-signaling- and cell-wall-organization-related genes and activates their expression via the removal of histone H3K27me3. Our findings reveal the crucial role of REF6-mediated histone H3K27 demethylation in the light-initiated seed germination process.

Seed dormancy is an adaptive trait in plants and determines the timing of germination [43]. During seed maturation, primary dormancy is induced in fresh seeds. Once the environmental conditions become favorable, the seeds break dormancy and germinate [43]. Two recent works suggested that REF6-mediated H3K27 demethylation plays an essential role in regulating seed dormancy [44,45]. It was reported that REF6 activates the expression of ABA catabolic genes *CYP707A1* and *CYP707A3*, leading to decreased ABA content and reduced seed dormancy [44]. REF6 also targets H3K27me3-marked genes in the endosperm and activates their expression to prevent seed dormancy [45]. These findings raised the speculation of whether the decreased seed germination of the *ref6-1* mutant under phyB-on conditions was caused by dormancy (Figure 1). In these works, freshly harvested or short-term-stored seeds were used to detect the germination trait [44,45]; however, in the present work, seeds stored for 30–35 days after harvesting were used for the germination assays. The wild-type and *ref6-1* seeds fully germinated 4 days after growing under WL conditions (Figure 1), revealing that the decreased germination of *ref6-1* may be independent of the dormancy trait. Consistently, during the preparation of the manuscript, the latest work published online demonstrated that REF6-mediated H3K27 demethylation is necessary for seed germination, and REF6 establishes an H3K27me3-depleted state that facilitates the activation of hormone-related and expansin genes important for germination [46]. Together, these findings confirm the critical role of REF6 in the regulation of seed germination. Notably, our work uncovered that REF6 specifically plays a positive role in the light-dependent seed germination process.

Previous genome-wide ChIP-seq analyses demonstrated that REF6 targets nearly 3000 genes in the Arabidopsis genome by directly recognizing the CTCTGYTY motif via its zinc-finger (ZnF) domains [37,40]. In the present work, transcriptome analysis revealed that REF6 activates 1628 genes and represses 1511 genes in imbibed seeds. It could be speculated that some of these genes are directly regulated while the other genes are indirectly regulated by REF6. Since REF6 is a histone H3K27 demethylase, it may generally act as the activator of target genes [37,38]. In this work, we found that REF6 associates with the CTCTGYTY motif of a subset of auxin-signaling- and cell-wall-organization-related genes and activates their expression via the removal of H3K27me3 (Figure 7, Figure 8 and Figure 9), which revealed that these genes are direct targets of REF6. Besides auxin-signaling- and cell-wall-organization-related genes, our RNA-seq analysis also showed that genes related to ABA and GA signaling are also regulated by REF6 (Figure 4 and Appendix A). Further qRT-PCR analysis confirmed that the expression of ABA biosynthesis genes *ABA1*, *ABA3* and *NCED6*, ABA signal transduction genes *ABI3* and *ABI4*, GA catabolic genes *GA2ox1*, *GA2ox2* and *GA2ox4*, and GA signal repressing genes *GAI*, *RGL2* and *RLG3* are up-regulated in *ref6-1* seeds (Appendix A). Although it is not clear how REF6 regulates these genes’ expression, the increased expression of these genes may lead to the inhibited seed germination of the *ref6* mutant. Consistent with the expression patterns of the GA- and ABA-signaling-related genes in the *ref6* mutant, our phenotypic analysis also reveals that REF6 may regulate phyB-dependent seed germination through GA and ABA signaling pathways (Figure 5 and Figure 6). Collectively, these findings reveal that REF6 may promote light-initiated seed germination by directly or indirectly regulating the expression of genes related to multiple plant hormonal signals in imbibed seeds.

Previous studies have reported the involvement of histone methylation at H3K4, H3K9 and H3K36 residues in light-regulated seed germination [26,27,28,29]; however, the role of the repressive H3K27me3 mark in this process remains unclear. Histone H3K27me3 is established by polycomb repressive complexes (PRCs) [47,48]. The core subunits of PRCs include Curly Leaf (CLF), Swinger (SWN), Fertilization Independent Endosperm (FIE), Embryonic Flower2 (EMF2) and Vernalization 2 (VRN2) [47,48]. Interestingly, a recent work reported that phyB directly interacts with Vernalization 5 (VRN5), a component of the PRC2 complex, to repress the expression of growth-promoting genes through the enrichment of H3K27me3 [49], which suggests the involvement of PRC-mediated H3K27me3 in phyB-regulated plant development. In future work, the role of these PRCs subunits in the light-regulated germination process could be investigated.

The histone H3K27me3 mark is removed by Jumonji domain-containing histone demethylases (JMJs). In Arabidopsis, the main H3K27me3 demethylases are Early Flowering 6 (ELF6/JMJ11), REF6/JMJ12, JMJ13 and JMJ30 [37,50,51,52]. These demethylases synergistically restrict the H3K27me3 level and repressive chromatin domains [50,51,52]. In the present work, the *REF6* knockout mutant displayed a decrease in the germination rate compared with the wild type under phyB-on conditions (Figure 1). It is not clear whether other JMJs play a role in light-initiated seed germination, although *elf6* and *jmj13* mutant seeds display no germination defects in normal growth conditions [48]. Further analysis of the phenotype of these JMJ mutants under light-dependent conditions will help to uncover their role in light-initiated seed germination.

## Figures and Tables

**Figure 1 cells-12-00295-f001:**
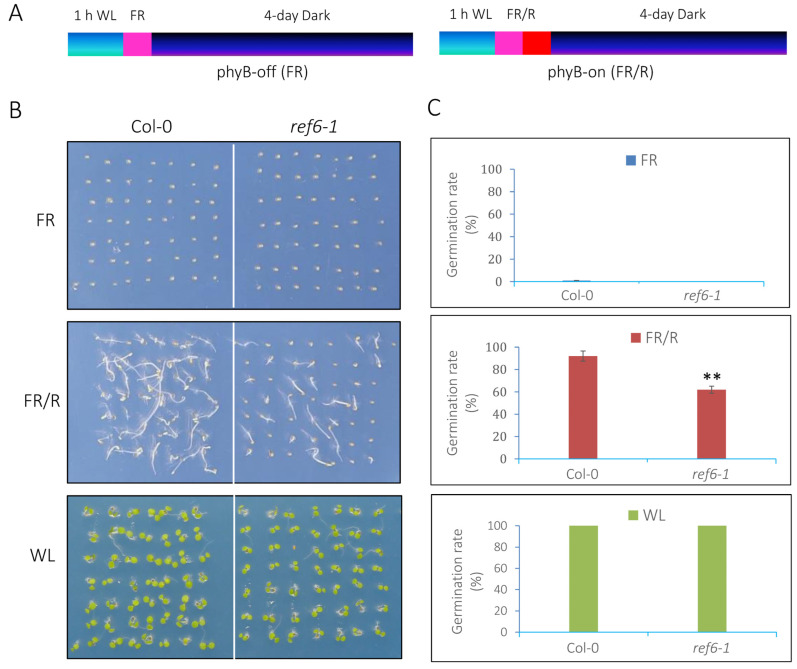
REF6 is a positive regulator of the phyB-dependent seed germination process. (**A**) Diagram displaying the phyB-inactivated (phyB-off, FR) and phyB-activated (phyB-on, FR/R) assays. (**B**) Germination patterns of Col-0 and *ref6-1* seeds under FR (phyB-off), FR/R (phyB-on) and WL (80 µmol m^−2^ s^−1^) conditions. (**C**) Statistical analysis of the germination rates of Col-0 and *ref6-1* seeds under FR, FR/R and WL conditions. Asterisks indicate significant differences (** *p* < 0.01, Student’s *t*-test). The experiments were performed in triplicate, and 50 seeds were scored for each sample.

**Figure 2 cells-12-00295-f002:**
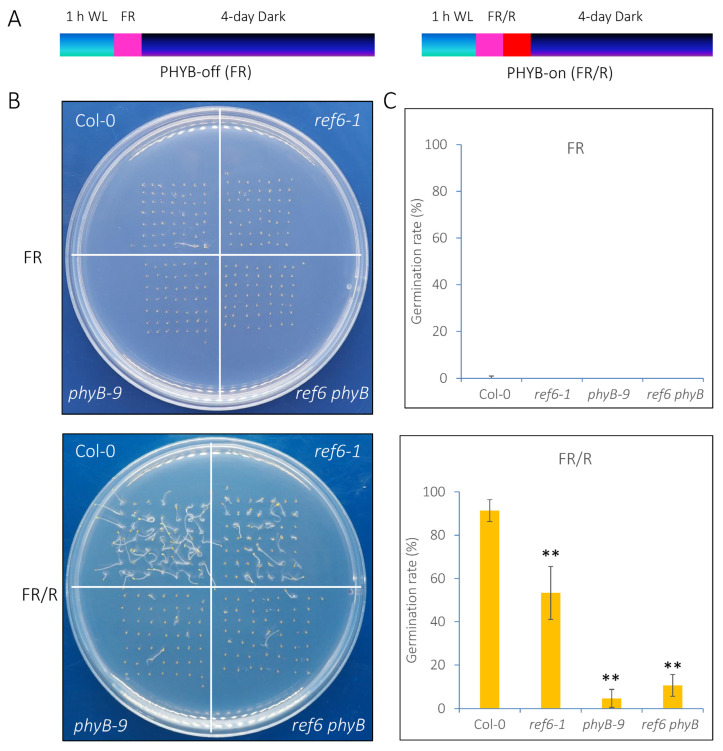
Genetic analysis of *REF6* and *phyB* in the phyB-dependent seed germination process. (**A**) Diagram displaying the phyB-inactivated (phyB-off, FR) and phyB-activated (phyB-on, FR/R) assays. (**B**) Germination patterns of Col-0, *ref6-1*, *phyB-9* and *ref6 phyB* seeds under FR (phyB-off) and FR/R (phyB-on) conditions. (**C**) Statistical analysis of the germination rates of Col-0, *ref6-1*, *phyB-9* and *ref6 phyB* seeds under FR and FR/R conditions. Asterisks indicate significant differences (** *p* < 0.01, Student’s *t*-test). The experiments were performed in triplicate, and 50 seeds were scored for each sample.

**Figure 3 cells-12-00295-f003:**
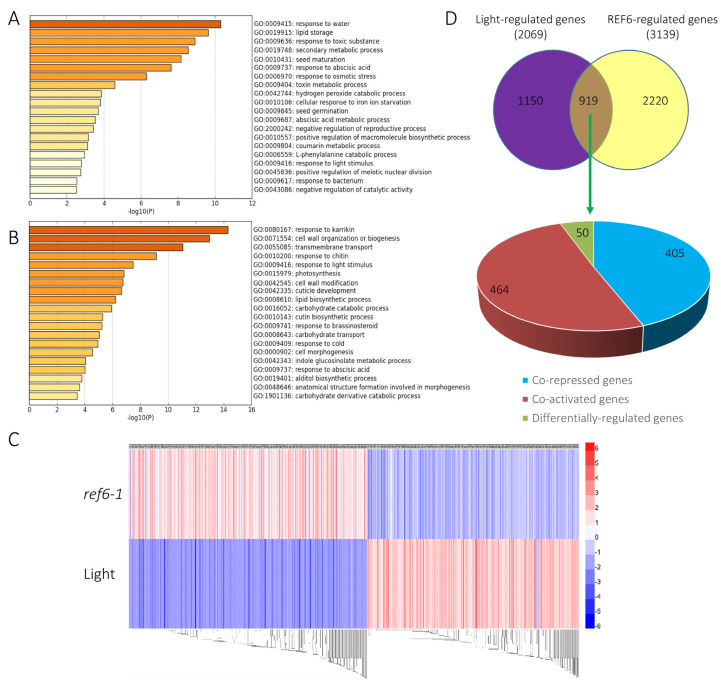
Genome-wide identification of REF6-regulated genes in imbibed seeds under FR/R conditions. (**A**) Chart of enriched ontology gene clusters repressed by REF6 (*p* < 0.05). (**B**) Chart of enriched ontology gene clusters activated by REF6 (*p* < 0.05). (**C**) Heatmap displaying the up- and down-regulated genes in *ref6-1* in comparison with the light-regulated genes as reported. (**D**) Venn diagram analysis of overlapping genes co-regulated by REF6 and light.

**Figure 4 cells-12-00295-f004:**
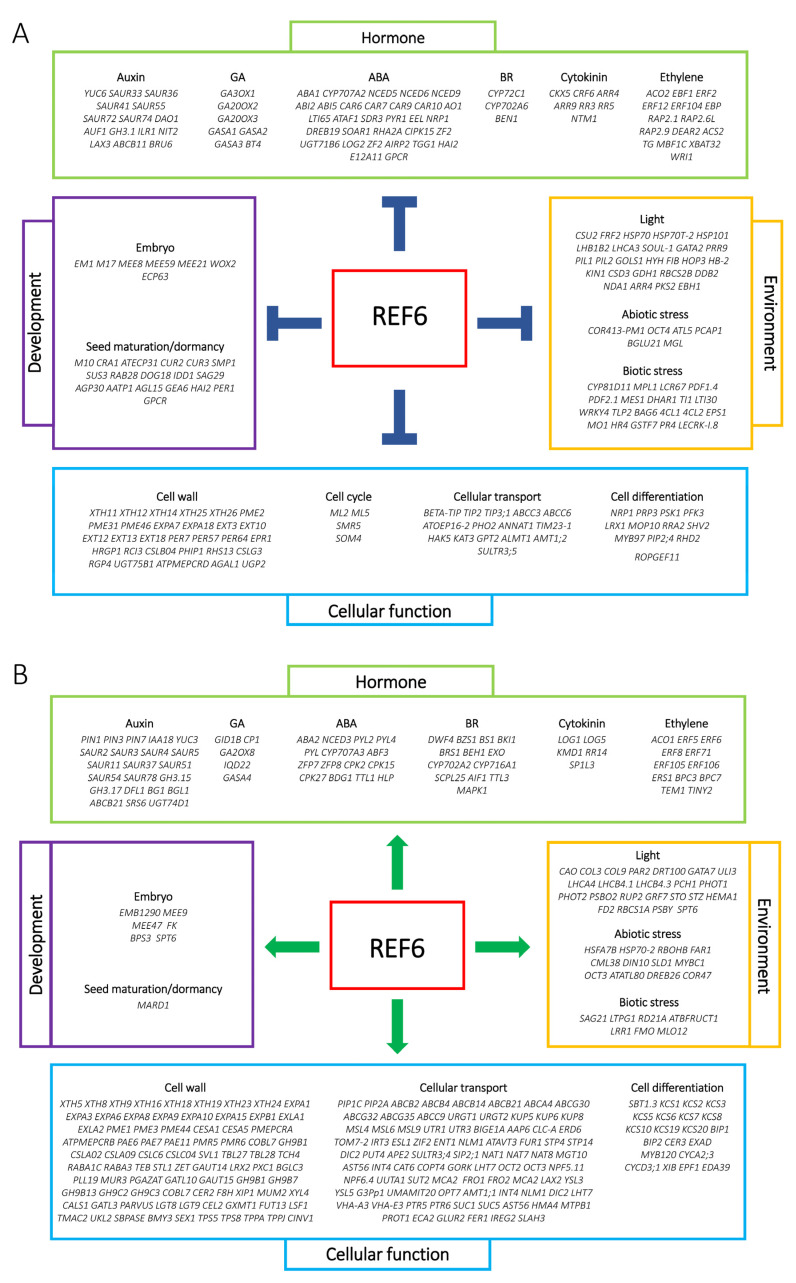
REF6 activated and repressed genes in imbibed seeds under FR/R conditions. (**A**) A chart displaying REF6-activated genes. (**B**) A chart displaying REF6-repressed genes.

**Figure 5 cells-12-00295-f005:**
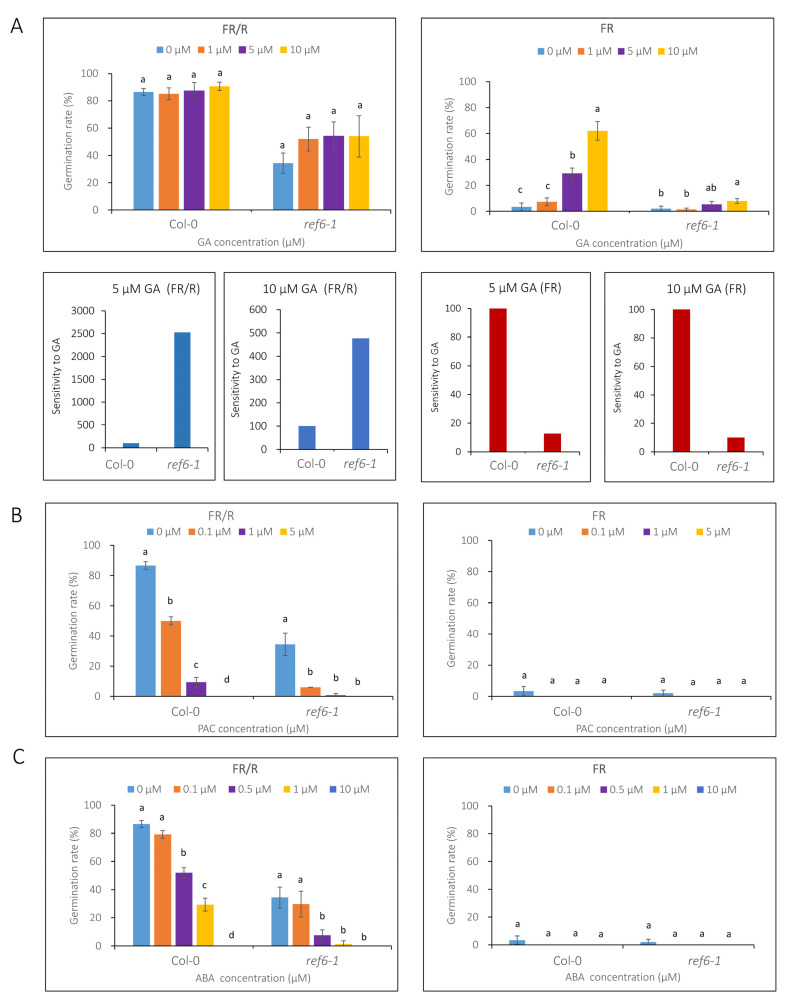
Effect of GA, PAC and ABA treatments on the seed germination of *ref6* mutant under phyB-dependent germination conditions. (**A**) Germination rates of *ref6-1* supplemented with 0, 1, 5 and 10 µM GA under phyB-on (FR/R) and phyB-off (FR) conditions. GA sensitivity was calculated as the ratio of the increase in germination rate of ref6 seeds in 5 and 10 µM GA and referenced to Col-0 seeds. (**B**) Germination rates of ref6-1 supplemented with 0, 0.1, 1 and 5 µM PAC under phyB-on and phyB-off conditions. (**C**) Germination rates of *ref6-1* supplemented with 0, 0.1, 0.5, 1 and 10 µM ABA under phyB-on and phyB-off conditions. Different concentrations of GA (GA3), PAC or ABA were added to the plates before treatments. At least 50 seeds were used for each sample and performed in triplicate. Values are shown as means ± SD (*n* = 3). The data were analyzed by one-way ANOVA. Different letters above bars indicate significant differences at *p* = 0.05.

**Figure 6 cells-12-00295-f006:**
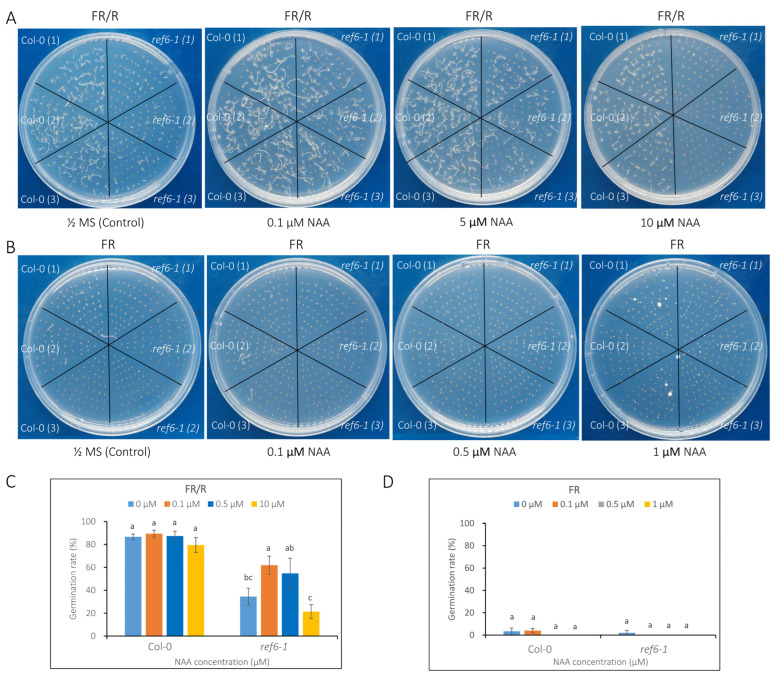
Effect of auxin (NAA) treatments on the seed germination of *ref6* mutant under phyB-dependent germination conditions. (**A**) Seed germination patterns of *ref6-1* supplemented with 0, 0.1, 5 and 10 µM NAA under phyB-on (FR/R) conditions. (**B**) Seed germination patterns of *ref6-1* supplemented with 0, 0.1, 0.5 and 1 µM NAA under phyB-off (FR) conditions. (**C**) Calculation of the germination rates of *ref6-1* supplemented with NAA under phyB-on conditions. (**D**) Calculation of the germination rates of *ref6-1* supplemented with NAA under phyB-off conditions. Different concentrations of NAA were added to the plates before treatment. At least 50 seeds were used for each sample, which was performed in triplicate. Values are shown as means ± SD (n = 3). The data were analyzed by one-way ANOVA. Different letters above bars indicate significant differences at *p* = 0.05.

**Figure 7 cells-12-00295-f007:**
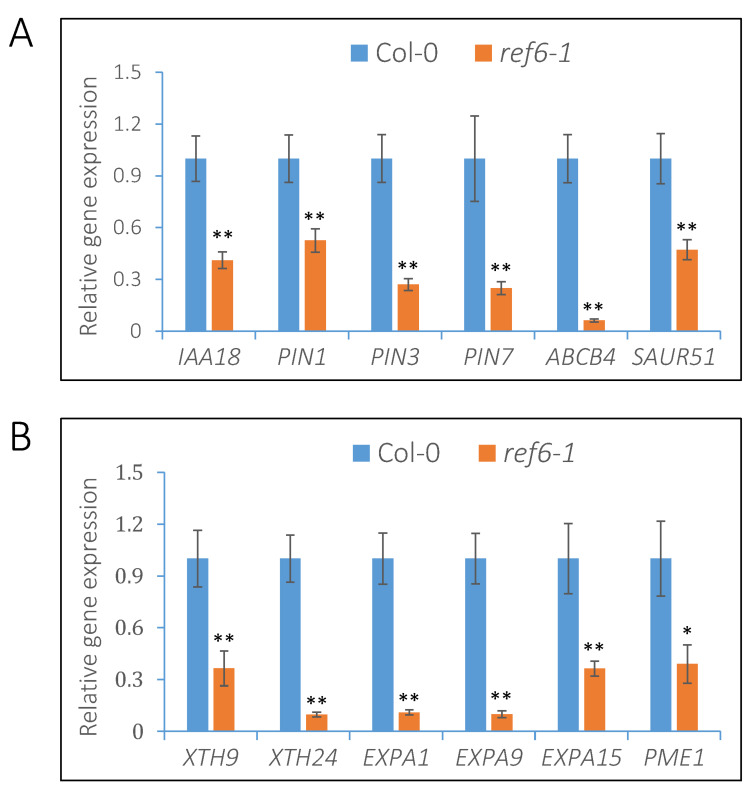
REF6 activates the expression of auxin-signaling- and cell-wall-loosening-related genes in imbibed seeds after FR/R irradiation. (**A**) qRT-PCR analysis of the expression levels of auxin-signaling-related genes in *ref6-1* mutant. (**B**) qRT-PCR analysis of the expression levels of cell-wall-organization-related genes in *ref6-1* seeds. *PP2A* was used as an internal control. Values are shown as means ± SD (* *p <* 0.05, ** *p <* 0.01, difference from Col-0, *n* = 3).

**Figure 8 cells-12-00295-f008:**
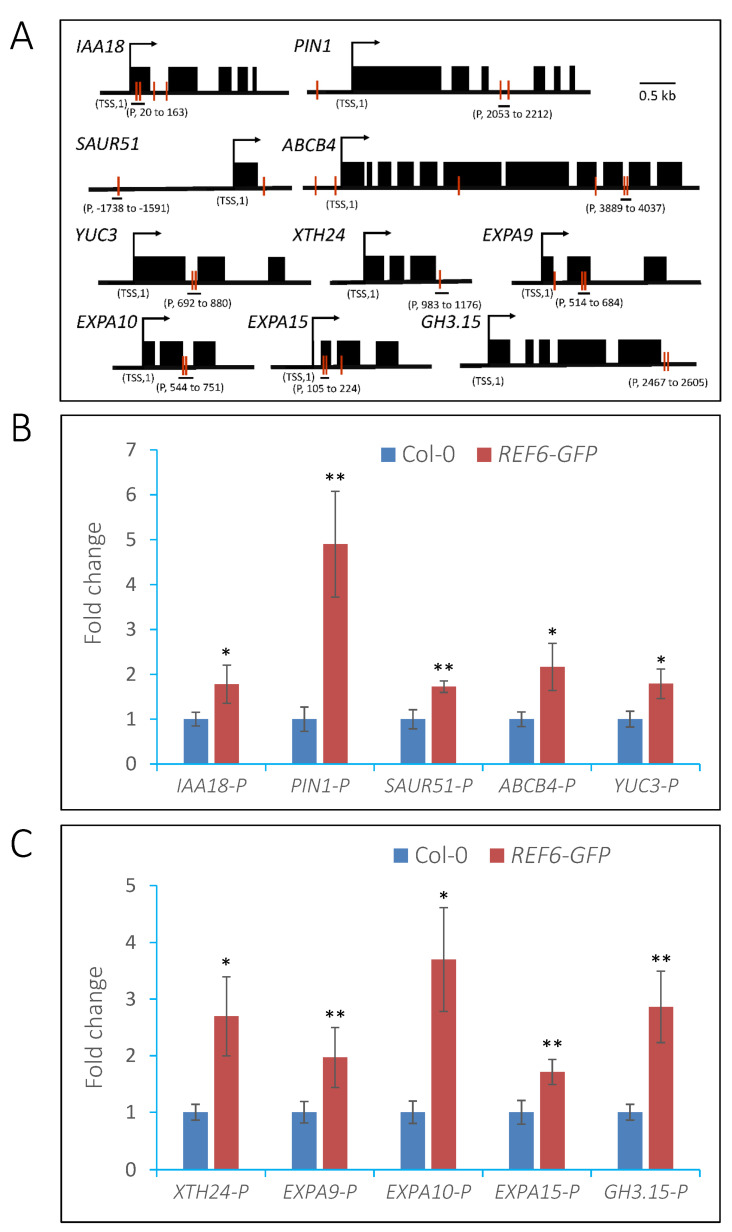
REF6 targets auxin-signaling and cell-wall-loosening-related genes under FR/R conditions. (**A**) Schematic diagram of the regions for ChIP analysis. The red lines indicate the CTCTGYTY motifs. TSS, transcription start site. The “P” regions were selected for analysis. (**B**) ChIP-qPCR analysis of enrichment of REF6 in auxin-signaling-related genes. (**C**) ChIP-qPCR analysis of enrichment of REF6 in cell-wall-loosening-related genes. After FR/R treatment, seeds were kept in the dark for 24 h and then crosslinked. The chromatin was extracted, sheared to an average of 500 bp and then immunoprecipitated with anti-GFP beads overnight. The DNA–protein complex was eluted and crosslinking was reversed. The immunoprecipitated DNA was extracted for qPCR analysis. The fold enrichment was normalized to the internal control *ACTIN2*. Values are shown as means ± SD (*t*-test, * *p* < 0.05, ** *p* < 0.01, *n* = 3).

**Figure 9 cells-12-00295-f009:**
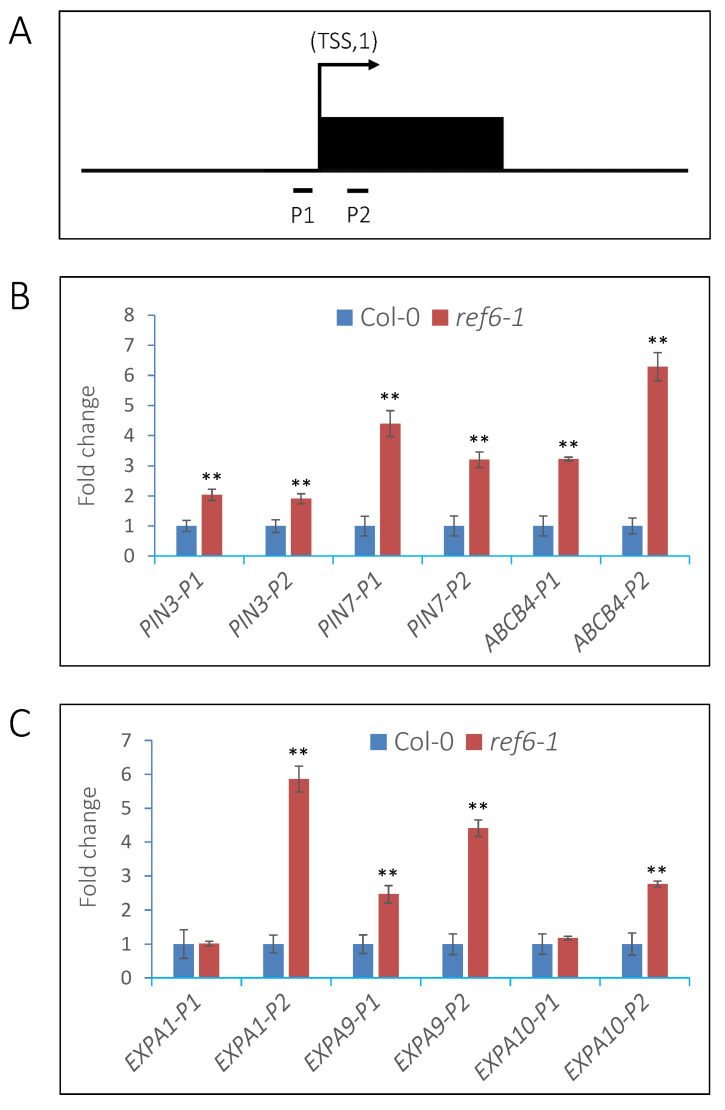
REF6 decreases the levels of H3K27me3 in the target genes in imbibed seeds under FR/R conditions. (**A**) Representative schematic diagram of the region for ChIP analysis. TSS indicates the transcription start site, and the black box indicates the first exon region. (**B**) ChIP-qPCR analysis of the H3K27me3 levels in auxin-signaling-related genes. (**C**) ChIP-qPCR analysis of the H3K27me3 levels of cell-wall-loosening-related genes. After FR/R treatment, seeds were kept in the dark for 24 h and then crosslinked. The chromatin was extracted, sheared to an average of 500 bp and then immunoprecipitated with anti-H3K27me3 antibody overnight. The DNA–protein complex was eluted and crosslinking was reversed. The immunoprecipitated DNA was extracted for qPCR analysis. *ACTIN2* was used as an internal control. Values are shown as means ± SD (*t*-test, ** *p* < 0.01, n = 3).

## Data Availability

Not applicable.

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
