# Peer review of "The Histone H3K27 Demethylase REF6 Is a Positive Regulator of Light-Initiated Seed Germination in Arabidopsis"

_cells, 2023, doi:10.3390/cells12020295_

Round 1
Reviewer 1 Report
Good work and well written. I have the following points for further improvement
1. please provide more details of the ChIP experiments in methods section for a reader to follow the experiments and replicate in their work.
2. the legends of Fig 5 and 6 also required to be explained in more detail.
3. check the spellings in line 70 "decorates", line 78 "mater" and line 271 "deceased"
Author Response
Good work and well written. I have the following points for further improvement
- please provide more details of the ChIP experiments in methods section for a reader to follow the experiments and replicate in their work.
Response: Thanks very much for the valuable suggestion. We have provided more details of the ChIP assay in the revised manuscript.
- the legends of Fig 5 and 6 also required to be explained in more detail.
Response: We have revised the legends of Fig.5 and 6 as suggested.
- check the spellings in line 70 "decorates", line 78 "mater" and line 271 "deceased"
Response: We have corrected the typos in revised manuscript.
Reviewer 2 Report
The manuscript by Wang et al comprises a very attention-grabbing investigation of the role of REF6 as a positive regulator of light-initiated seed germination.
The main flaw of the study is that no methodology is reported on the approach used by the authors to support the direct regulation of REF6 with hundreds of genes. Using mutants is not enough since it cannot exclude an indirect link. To this end, the authors should at least carry out a network analysis to estimate the level of correlation among genes.
As a minor flaw, I would suggest improving the resolution of figure 1. Moreover, the caption of figure 1 should not report the germination protocol. The author can refer to the dedicated section or use coherent info-graphics.
Author Response
The manuscript by Wang et al comprises a very attention-grabbing investigation of the role of REF6 as a positive regulator of light-initiated seed germination.
The main flaw of the study is that no methodology is reported on the approach used by the authors to support the direct regulation of REF6 with hundreds of genes. Using mutants is not enough since it cannot exclude an indirect link. To this end, the authors should at least carry out a network analysis to estimate the level of correlation among genes.
Response: We totally agree the opinion that we have no global data to support the direct regulation of REF6 with hundreds of genes, such as ChIP-seq analysis. However, we focused on the auxin signal and cell wall organization related genes which closed related to seed germination. We have proved that these genes are direct targets of REF6 by ChIP-qPCR assays. In the furture, genome-wide analysis of REF6-bound genes is needed to elucidate its role in light-dependent seed germination process.
As a minor flaw, I would suggest improving the resolution of figure 1. Moreover, the caption of figure 1 should not report the germination protocol. The author can refer to the dedicated section or use coherent info-graphics.
Response: We have improved the resolution of figure 1, and the germination protocol has been removed from the caption of figure 1.
Reviewer 3 Report
This manuscript describes how REF6 regulates phyB-dependent germination controlling the methylation and thus the expression of different genes involved in seed germination. The conclusions are based mainly on molecular experiments, and I think that some genetics is missing. In particular, I suggest the following experiments:
- To better understand the involvement of REF6 in the phyB-dependent pathway, I believe that a study of the epistasis between phyB and ref6 mutants is necessary. Hence, I would like to see the germination in FR/R of the phybref6 double mutant and compare it with the single phyB and ref6 mutants.
- As a control, I would like to see the germination of the transgenic line REF6-GFP in FR/R used in the ChIP-seq experiment, because I expect complementation of the ref6 mutant phenotype.
- It seems that REF6 controls seed germination by acting on hormone-related genes. To better understand how REF6 is involved in hormone-mediated seed germination, I suggest studying the ref6 germination phenotype in a dose-response experiment with at least ABA, GA, Pacobutrazol and Auxin.
Other minor comments are the following:
The figure 3, the authors show the genes deregulated in the ref6 mutant. It would be better to have heatmaps of the main groups of genes, in order to highlight the amplitude of the gene expression.
REF6 regulates seed germination, but how is it regulated during germination? Are its protein and/or expression levels regulated by light and/or imbibition? It would be nice to show gene expression and/or protein levels of REF6 in FR vs FR/R treated seeds and during imbibition in white light-treated seeds.
The results of the ChIP experiment shown in figure 5 are described as Fold enrichment and in figure 6 as fold change, there is a difference in the calculation, or is it just a mistake?
To help the reproducibility of the results, it is helpful to have detailed materials and methods. In particular, I could not find information on the calculation of the RT-qPCR and ChIP-qPCR results, hence I suggest adding this information.
Author Response
This manuscript describes how REF6 regulates phyB-dependent germination controlling the methylation and thus the expression of different genes involved in seed germination. The conclusions are based mainly on molecular experiments, and I think that some genetics is missing. In particular, I suggest the following experiments:
- To better understand the involvement of REF6 in the phyB-dependent pathway, I believe that a study of the epistasis between phyBandref6 mutants is necessary. Hence, I would like to see the germination in FR/R of the phybref6 double mutant and compare it with the single phyB and ref6 mutants.
Response: We totally agree the opinion that more genetic works should be performed to verify the role of REF6 in the phyB-dependent germination pathway. Actually, we have already generated ref6 phyB double mutant in hand. Recently, we have analyzed the phenotype of ref6 phyB double mutant by light-dependent seed germination assays. We have added these data in the revised manuscript (Figure 2).
- As a control, I would like to see the germination of the transgenic line REF6-GFP in FR/R used in the ChIP-seq experiment, because I expect complementation of the ref6mutant phenotype.
Response: Expression of REF6-GFP in ref6-1 background could fully rescue the germination defect of ref6-1 seeds under FR/R condition, which suggested that REF6 is functional in vivo. We have added this data in revised manuscript (Supplemental Figure S2).
It seems that REF6 controls seed germination by acting on hormone-related genes. To better understand how REF6 is involved in hormone-mediated seed germination, I suggest studying the ref6 germination phenotype in a dose-response experiment with at least ABA, GA, Pacobutrazol and Auxin.
Response: We totally agree with the idea that REF6 controls seed germination via hormone signal pathways. We have detected the seed germination rates of ref6-1 mutants after treated with different levels of ABA, GA, PAC and auxin under PHYB-on and PHYB-off conditions. Our phenotypic analyses suggested that REF6 regulates PHYB-dependent seed germination not only through the GA and ABA signal processes, but also greatly depends on the auxin pathway (Figure 5 and 6).
Other minor comments are the following:
- The figure 3, the authors show the genes deregulated in the ref6mutant. It would be better to have heatmaps of the main groups of genes, in order to highlight the amplitude of the gene expression.
Response: Good idea, we have added the heatmap data in the revised manuscript (Figure 3).
- REF6 regulates seed germination, but how is it regulated during germination? Are its protein and/or expression levels regulated by light and/or imbibition? It would be nice to show gene expression and/or protein levels of REF6 in FR vs FR/R treated seeds and during imbibition in white light-treated seeds.
Response: We have detected gene expression pattern of REF6 in imbibed seeds under both FR and FR/R conditions. It was shown that the expression of REF6 is up-regulated 3 and 6 h after FR/R treatment, whereas down-regulated 3, 6, 12 and 24 h after FR treatment. However, the protein levels of REF6-GFP in imbibed seeds are difficult to be detected. We have added the gene expression data of REF6 in the revised manuscript (Supplemental Figure S1).
- The results of the ChIP experiment shown in figure 5 are described as Fold enrichment and in figure 6 as fold change, there is a difference in the calculation, or is it just a mistake?
Response: Indeed it’s a mistake. We have corrected to fold change.
- To help the reproducibility of the results, it is helpful to have detailed materials and methods. In particular, I could not find information on the calculation of the RT-qPCR and ChIP-qPCR results, hence I suggest adding this information.
Response: We have added the detailed information of these methods in the revised manuscript.
Round 2
Reviewer 2 Report
The authors amended the MS according to the raised criticisms.
Author Response
Thank you very much for your comments.
Reviewer 3 Report
I very much appreciate the genetic experiments carried out by the authors.
Based on the experiments, the authors write from line 300:” Above data revealed that REF6 may promote PHYB-activated seed germination largely via auxin signal pathway. Collectively, our phenotypic analyses suggested that REF6 regulates PHYB-dependent seed germination not only through the GA and ABA signaling pathways but also greatly depends on the auxin pathway
But, I have the following doubts:
11) Does the phyB-dependent germination really require auxin? The literature is clear about GA and ABA, but I could not find any information about auxin. Can the authors cite relevant literature about phyB and auxin in seed germination?
22) The experiment shown in figure 6 suggests that ref6 mutants are sensitive to auxin, meaning that REF6-dependent germination needs auxin, but it does not show that phyB-dependent germination depends on auxin. I believe that further analyses are necessary to say that.
32) The experiment in Figure 5 suggests that the GA treatment does not rescue ref6 germination, but does it respond to GA? I think that some statistics to show the significance of ref6 response to GA could help the interpretation.
Hence, I would suggest: a) adding statistical analysis to the germination experiments; b) changing the conclusion of the experiment from line 300 in this way: “Above data revealed that REF6 may promote seed germination largely via auxin signal pathway. Collectively, our phenotypic analyses suggested that REF6 regulates seed germination not only through the GA and ABA signaling pathways but also greatly depends on the auxin pathway.” And also change the abstract: “Phenotypic analyses indicated that REF6 regulates seed germination not only through GA (gibberellin) and ABA (abscisic acid) processes, but it also depends on the auxin signal pathway.
Last minor comment, sorry that I did not notice this during the first revision, but the nomenclature for photoreceptors wants phytochrome B written always minuscule phyB. PHYB is referred to as the protein without the chromophore and this is not the case. Hence, please change PHYB in phyB in the text.
Author Response
I very much appreciate the genetic experiments carried out by the authors.
Based on the experiments, the authors write from line 300:” Above data revealed that REF6 may promote PHYB-activated seed germination largely via auxin signal pathway. Collectively, our phenotypic analyses suggested that REF6 regulates PHYB-dependent seed germination not only through the GA and ABA signaling pathways but also greatly depends on the auxin pathway
But, I have the following doubts:
11) Does the phyB-dependent germination really require auxin? The literature is clear about GA and ABA, but I could not find any information about auxin. Can the authors cite relevant literature about phyB and auxin in seed germination?
Response: We also didn’t found any information about the role of auxin in phyB-dependent germination, but it’s an interesting question whether PHYB-dependent germination really requires auxin.
22) The experiment shown in figure 6 suggests that ref6 mutants are sensitive to auxin, meaning that REF6-dependent germination needs auxin, but it does not show that phyB-dependent germination depends on auxin. I believe that further analyses are necessary to say that.
Response: We totally agree this opinion. We have changed the related description in the text.
32) The experiment in Figure 5 suggests that the GA treatment does not rescue ref6 germination, but does it respond to GA? I think that some statistics to show the significance of ref6 response to GA could help the interpretation.
Hence, I would suggest: a) adding statistical analysis to the germination experiments; b) changing the conclusion of the experiment from line 300 in this way: “Above data revealed that REF6 may promote seed germination largely via auxin signal pathway. Collectively, our phenotypic analyses suggested that REF6 regulates seed germination not only through the GA and ABA signaling pathways but also greatly depends on the auxin pathway.” And also change the abstract: “Phenotypic analyses indicated that REF6 regulates seed germination not only through GA (gibberellin) and ABA (abscisic acid) processes, but it also depends on the auxin signal pathway.
Response: Thanks very much for the comments. We have added statistical analysis to analyze the sensitivity of ref6 mutant in response to GA. It was found that ref6 seeds are more sensitive to GA under phyB-on conditions, while less sensitive to GA under phyB-off conditions compared to Col-0 (Figure 5). We have changed the description as suggested.
Last minor comment, sorry that I did not notice this during the first revision, but the nomenclature for photoreceptors wants phytochrome B written always minuscule phyB. PHYB is referred to as the protein without the chromophore and this is not the case. Hence, please change PHYB in phyB in the text.
Response: Thanks very much for this information and suggestion. We have changed PHYB to phyB throughout the manuscript.